# Water Use Efficiency Spatiotemporal Change and Its Driving Analysis on the Mongolian Plateau

**DOI:** 10.3390/s25072214

**Published:** 2025-04-01

**Authors:** Gesi Tang, Yulong Bao, Changqing Sun, Mei Yong, Byambakhuu Gantumur, Rentsenduger Boldbayar, Yuhai Bao

**Affiliations:** 1College of Geographical Science, Inner Mongolia Normal University, Hohhot 010022, China; 20204019045@mails.imnu.edu.cn (G.T.); 21d1num0158@stud.num.edu.mn (C.S.); yongmei2012@imnu.edu.cn (M.Y.); baoyuhai@imnu.edu.cn (Y.B.); 2Key Laboratory of Mongolian Plateau Geographical Research, Inner Mongolia Autonomous Region, Hohhot 010022, China; 3Department of Geography, School of Arts and Sciences, National University of Mongolia, Ulaanbaatar 14200, Mongolia; byambakhuu@num.edu.mn; 4Laboratory of Geoinformatics (GEO-iLAB), Graduate School, National University of Mongolia, Ulaanbaatar 14200, Mongolia; 5Division of GIS and Remote Sensing, Institute of Geography and Geoecology, Mongolian Academy of Sciences, Ulaanbaatar 15170, Mongolia; brentsenduger@igsnrr.ac.cn

**Keywords:** water use efficiency, vegetation phenology, remote sensing, Mongolian Plateau, evapotranspiration

## Abstract

Water use efficiency (WUE) connects two key processes in terrestrial ecosystems: the carbon and water cycles. Thus, it is important to evaluate temporal and spatial changes in WUE over a prolonged period. The spatiotemporal variation characteristics of the WUE in the Mongolian Plateau from 1982 to 2018 were analyzed based on the net primary productivity (NPP), evapotranspiration (ET), temperature, precipitation, and soil moisture. In this study, we used remote sensing data and various statistical methods to evaluate the spatiotemporal patterns of water use efficiency and their potential influencing factors on the Mongolian Plateau from 1982 to 2018. In total, 27.02% of the region witnessed a significant decline in the annual WUE over the 37 years. Two abnormal surges in the WUE_Season_ (April–October) were detected, from 1997 to 1998 and from 2007 to 2009. The trend in the annual WUE in some broadleaf forest areas in the middle and northeast of the Mongolian Plateau reversed from the original decreasing trend to an increasing trend. WUE has shown strong resilience in previous analytical studies, whereas the WUE in the artificial vegetation area in the middle of the Mongolian Plateau showed weak resilience. WUE had a significant positive correlation with precipitation, soil moisture, and the drought severity index (DSI) but a weak correlation with temperature. WUE had strong resistance to abnormal water disturbances; however, its resistance to the effects of temperature and DSI anomalies was weak. The degree of interpretation of vegetation changes for WUE was higher than that for meteorological factors, and WUE showed weak resistance to normalized difference vegetation index (NDVI) disturbances. Delaying the start of the vegetation growing season had an increasing effect on WUE, and the interaction between phenological and meteorological vegetation factors had a non-linear enhancing effect on WUE. Human activities have contributed significantly to the increase in WUE in the eastern, central, and southern regions of the Mongolian Plateau. These results provide a reference for the study of the carbon–water cycle in the Mongolian Plateau.

## 1. Introduction

Global warming, caused by human activities, is becoming increasingly serious, and the global water cycle is intensifying [1,2]. Water shortages are a serious problem in arid and semiarid areas [3,4]. Owing to the progress of civilization and the intensification of human activity, speeding up the greenhouse effect, all countries experienced a water crisis at the beginning of the 21st century. This crisis was particularly prominent in arid and semiarid areas, such as the Mongolian Plateau. As an important indicator, the water use efficiency (WUE, gC/mm·m^2^) can be used to understand the relationship between water and the carbon cycle in ecosystems, as it reveals the balance between carbon storage and water loss [5,6]. Determining the spatiotemporal variation characteristics and control mechanisms of the WUE of regional-scale ecosystems helps to evaluate and predict the response of an ecosystem’s carbon–water cycle to global changes.

At the ecosystem scale, water use efficiency is generally defined as the ratio of carbon storage (such as net primary production, NPP, or gross primary production, GPP) to water loss (such as transmission, T, or evapotranspiration, ET) [7,8]. On a large regional scale, a model integrated with remote sensing data is usually used to calculate the productivity (GPP, NPP) and ET to estimate the WUE, which is calculated as the ratio of the two (defined as WUE = NPP/ET in this study) [9,10]. Recently, research on WUE has attracted much attention, especially the correlation between water use efficiency and meteorological variables and technology, seeking to improve vegetation’s WUE [10,11,12,13]. Previous studies have shown that the rise in carbon dioxide concentrations in many areas has led to an increase in the utilization rate of water resources [14,15,16]. Simultaneously, owing to the increase in the aerosol load in global terrestrial ecosystems, the WUE continues to increase every year [8]. Low precipitation in the arid and semiarid areas of the Mongolian Plateau has led to perennial droughts, whereas the intensification of climate change has rendered already fragile ecosystems more vulnerable to damage. Therefore, it is imperative to study the long-term spatiotemporal dynamics of water resource utilization efficiency in the Mongolian Plateau and explore the impact of the unique climate of the Mongolian Plateau on this efficiency.

However, flux observation stations and model construction are unsuitable for large-scale WUE research. For example, numerous scholars have conducted research on WUE using flux observation data; however, these studies are typically limited to small spatial scales [17,18,19]. Additionally, some researchers have employed models to simulate the GPP and ET to investigate long-term and large-scale WUE [20]. However, these studies are often accompanied by inherent uncertainties within the models themselves [21,22,23]. Fortunately, low-cost satellite remote sensing can be used to monitor the biological characteristics of various ecosystems. The latest-generation GIMMS NDVI (NDVI3 g v1.0) has been proven to have the highest time consistency and provides the NDVI with the longest time scale (>30 years) [24,25,26]. Moderate-resolution imaging spectrometry (MODIS) provides the mod13 NDVI product and has a high spatiotemporal resolution. This instrument has greatly promoted research on the regional-scale WUE, and, in the past, remote sensing data have been effectively used to study water use efficiency [27,28]. However, research on the water resource utilization efficiency at the ecosystem scale is still in the primary stage, and the process of quantifying the water resource utilization efficiency using remote sensing data remains insufficient. In the past, most of the factors considered to affect WUE were meteorological factors.

However, WUE is affected not only by the climatic environment but also by vegetation changes. For example, changes in the vegetation growth season may affect the quality of vegetation growth and the carbon storage capacity, thus directly affecting the WUE [29]. At the same time, the degree of interpretation of GPP regarding WUE anomalies in Inner Mongolia is higher than that for ET; however, the relationship between WUE anomalies and vegetation greenness and meteorological factor anomalies is still unclear [30]. Phenology refers to the study of natural events such as growth and reproduction in vegetation, animals, and other organisms, which may occur earlier or later in response to climate change. Rising temperatures often cause phenomena like vegetation blooming and animal activity to occur earlier, affecting ecosystems and agricultural production. Phenology changes are an important indicator of global warming, reflecting the impact of climate change on nature, and they have become a key metric in global climate monitoring [31,32,33,34]. Most previous studies have shown that carbon storage is affected by the temperature and humidity [35,36,37,38,39]. Changes in vegetation phenology also affect carbon storage and sinks. For example, an earlier ‘turning green’ period may cause plant leaves to be larger, intercepting more light in the canopy, thereby improving the light efficiency and increasing the spring GPP and vegetation transpiration [40]. Some studies have shown that changes in the phenological parameters of vegetation affect carbon assimilation [41,42,43]. Thus, these parameters may be able to indicate the impact of vegetation changes on WUE. To explore the influence of phenological parameters on water use efficiency, we considered the effects of the vegetation phenology on both NPP and ET. In addition, due to the complexity of the ecosystem, the impact of changes in the growing season on WUE may not be consistent across different regions.

Most previous studies on WUE in the Mongolian Plateau have focused on the relationship between WUE and meteorological factors over the past 20 years [30,44]. Studies on WUE over long periods and trend analyses of various models are limited, and research on the synergistic effects of vegetation and meteorological factors and the impact of vegetation and climate anomalies on the WUE in the Mongolian Plateau is still insufficient. In addition, changes in vegetation phenology due to climate change may lead to changes in NPP and ET, indirectly affecting WUE. Thus, it is of great significance to study the impact of the vegetation phenology on WUE and the potential mechanisms underlying this impact. There are few studies on the restoration ability of an ecosystem’s WUE that consider past anomalies and their resistance to abnormal changes in meteorological factors and vegetation greenness. In addition to natural factors, the effects of human interference on ecosystems should be considered. Studying the synergistic effects of meteorological factors and vegetation on the water use efficiency in the Mongolian Plateau is of great significance for a more comprehensive understanding of the mechanism of water use efficiency changes.

## 2. Materials and Research Methods

### 2.1. Study Area

The Mongolian Plateau (87°43′–126°04′ E, 37°22′–53°23′ N) is part of the Central Asian Plateau. It has both arid and semiarid areas, experiences a typical continental climate, and includes the entire territory of Mongolia and the Inner Mongolia Autonomous Region of China [45]. The Mongolian Plateau is bordered by high-altitude mountains to the north and is adjacent to the Gobi Desert to the south [46,47]. Affected by the climate, the annual precipitation level in the Mongolian Plateau is low, and droughts are frequent. The average annual precipitation in most areas is 200 mm, which occurs mainly from June to August. It is windy in winter and spring, while the average temperature in January is −26 °C and that in July is 17 °C. The plateau also has complex ecological zones, including meadow steppes, typical steppes, alpine steppes, desert steppes, coniferous forests, broadleaf forests, and the Gobi Desert; see Figure 1 for details. Vegetation data were obtained from the National Atlas of Mongolia, and an IMG vegetation map with a scale of 1:1,000,000 was rasterized to a resolution of 0.083° [48,49].

### 2.2. Data Sources

#### 2.2.1. NPP and ET Dataset

The net primary productivity data were taken from GLASS NPP products developed by Yuan et al., with a spatial resolution of 5 km (0.05°) and a temporal resolution of 8 days [50,51]. This product has the advantages of high precision, a high space–time resolution, and a long time series (from 1982 to 2018). The dataset can be accessed at (https://glass.hku.hk/, accessed on 18 March 2024).

The total evapotranspiration of the ecosystem was determined from the global land surface characteristic parameter product named evapotranspiration AVHRR ET (5 km), developed by Yao et al., which has a time resolution of 8 days and contains data for 1982–2018 [52]. The Bayesian average method was used on the AVHRR and MERRA2 data to integrate five evapotranspiration algorithms and obtain the dataset [53,54]. This product has a long duration, a high resolution, and high precision, thus providing a reliable basis for the study of global environmental change. The dataset was accessed from the National Earth System Science Data Center, National Science and Technology Infrastructure of China (http://www.geodata.cn, accessed on 18 March 2024).

#### 2.2.2. Normalized Difference Vegetation Index (NDVI) Dataset

The GIMMS NDVI3g v1.0 dataset (1981–2015) was obtained from NASA’s NOAA-AVHRR data, with a spatial resolution of 0.083° and a temporal resolution of 15 days. The monthly NDVI was derived using the maximum synthesis method. Supplementary MODIS MOD13A3 data (2001–2018) with a 1 km resolution were also used. We aligned both datasets by resampling them to 0.083° and created a regression model using 2000–2009 data, validating it with those for 2010–2015. The fused GIMMS-MODIS NDVI data (2010–2015) showed a strong correlation with the original GIMMS data, allowing reliable expansion to 2016–2018. For the GLASS NPP and ET data, we resampled them to 5 km using bilinear interpolation, resulting in a 1982–2018 GIMMS-MODIS NDVI dataset with a 5 km spatial resolution. GIMMS: (https://www.nasa.gov/nex/, accessed on 18 March 2024). MODIS: (https://appeears.earthdatacloud.nasa.gov/, accessed on 18 March 2024).

#### 2.2.3. Meteorological Dataset

We used the total precipitation (PRE), surface temperature (TEM), soil moisture (SW), and potential evapotranspiration (PET) in this study. All data were monthly-scale data with a 0.1° spatial resolution, obtained from ERA5-LAND from 1982 to 2018. The data were obtained from (https://cds.climate.copernicus.eu/cdsapp#!/dataset/reanalysis-era5-land-monthly-means?tab=overview, accessed on 18 March 2024).

Drought severity indices (DSIs) can provide more information about water stress [55]. The DSI was estimated using extended GIMMS MODIS NDVI, AVHRR ET, and ERA5-LAND PET data over 37 years.

#### 2.2.4. WUE Dataset

The WUE was calculated as the ratio of the GLASS NPP to the AVHRR ET. Since the focus of this study was the land water use efficiency in vegetation areas, we removed non-vegetation areas (average NDVI < 0.1), ET, and NPP before the inversion of the water use efficiency. Therefore, the final WUE also excluded data from non-vegetated areas.

#### 2.2.5. Inversion of Vegetation Phenological Parameters

After discarding non-vegetation areas with annual average NDVI values < 0.1 [56], we used the expanded monthly GIMMS-MODIS NDVI (1982–2018) to extract the phenological parameters (SOS and EOS) of vegetation on the Mongolian Plateau using the logistic curve maximum curvature method. This method was considered effective in previous studies [57,58]. First, a logistic model was used to fit the daily NDVI after accumulated interpolation. The specific calculations were as follows:(1)$$y\left(t\right)=\frac{c}{1+e^{a+bt}}+d$$
where y(t) is the cumulative NDVI fitting value of t in a year; a and b are model parameters; d is the daily minimum NDVI; and c + d is the maximum cumulative NDVI. Then, the curvature of the fitting curve at each time t was calculated; the time corresponding to the first maximum curvature was defined as the SOS, and the time corresponding to the last local maximum curvature was defined as the EOS.(2)$$\mathrm{curvature}=\frac{da}{ds}=-\frac{b^{2}ce^{a+bt}\left(1-e^{a+bt}\right){\left(1+e^{a+bt}\right)}^{2}}{{\left[{\left(1+e^{a+bt}\right)}^{2}+{\left(bce^{a+bt}\right)}^{2}\right]}^{\frac{3}{2}}}$$

### 2.3. Research Methods

#### 2.3.1. Theil–Sen Trend Analysis and Mann–Kendall Test

Theil–Sen trend analysis and the Mann–Kendall test are effective in assessing trends in long-time-series data. Combining these methods has the following advantages. They do not require a certain data distribution and can tolerate a certain degree of data error [59]. Additionally, the Mann–Kendall test was used to test the significance of the linear trends [60,61,62]. Please refer to the attachment for the calculation processes of the Theil–Sen trend analysis and MK test. In this study, we analyzed the trends in NPP, ET, WUE, and DSI from 1982 to 2018.

#### 2.3.2. Breaks for Additive Season and Trend (BFAST) Package

The Breaks for Additive Season and Trend method was originally proposed by Verbesselt to detect interference in the time series of remote sensing data [63]. The BFAST algorithm is widely used in the fields of meteorology, climatology, economic development, and vegetation dynamics [64,65,66]. It uses parameters and seasonal trend models that fit piecewise linear trends [67]. Compared to traditional interannual linear regression, BFAST can express more trend details and detect mutation breakpoints. A breakpoint refers to a point in time-series data where a variable undergoes a significant change, usually caused by external shocks or internal mechanisms. This formula is defined as follows:(3)$${WUE}_{t}=Trendt+Seasonalt+Remaindert,\;t=1,2,3\ldots \ldots n$$(4)Trendt=αi+βit(5)$$Seasonalt=\sum_{k-1}^{j}\alpha_{j,k}sin\left(\frac{2\pi kt}{f}\right)+\delta_{j,k}$$
where WUEt, Trendt, Seasonalt, and Remainder are the 8-day WUE_Season_ from 1982 to 2018, the trend component, the seasonal component, and the remaining component, respectively. The trend is piecewise linear with the gradient of a specific road section and an intercept of l + 1 different parts. Therefore, there are l breakpoints, τi − 1, … τl, where i = 1, 2, 3 …… L, and, by definition, τ0 = 0 and τl + 1 = n.

The magnitude of breakpoint L and the intercept of Trendt between αi and βi can be used to estimate ti − 1 and ti. t0 = 0 is defined for tj < t ≤ tj + 1 to adapt the frequency, where j represents the time of occurrence of the breakpoint, j = 1, 2, 3 … l, and k represents the number of harmonic terms. The total number of breakpoints is l, and αj,k and δj,k represent the segment-specific amplitude and phase, respectively. In this study, the WUE, NPP, and ET values during the growing season of the Mongolian Plateau (April–October) from 1982 to 2018 were used for the breakpoint analysis.

#### 2.3.3. WUE Change Point Detection

The change point detection tool in the ArcGIS pro3.0 software was used to detect the change points of the annual WUE. This tool enables us to check changes in the slope (linear trend) of discrete variables. Change point detection is relatively similar to time-series anomaly detection but differs in several important aspects. When a model is changed to a new model (such as an average value change), change point detection identifies the time step, whereas outlier detection identifies time steps that significantly deviate from a single model. The former highlights continuous changes, whereas the latter highlights short-term abnormalities. In this study, we used the slope change to detect interannual WUE trend change points from 1982 to 2018. The optimal change point detection with the linear computing cost (PELT) option uses the pruning precision linear time [68,69] algorithm to estimate the number of changes and time steps. The reduction in the subdivision cost must be greater than the additional penalty value determined by the sensitivity parameter. Based on the characteristics of the constraints, a penalty function was constructed and added to the objective function to establish an unconstrained problem using an empirical formula. If the cost reduction is less than the increased penalty value, the penalty cost will improve, and the time step is not defined as a change point. If the sensitivity value is between 0 and 1, more detection points are obtained with high sensitivity. The penalty formula is as follows:(6)$$\mathrm{Penalty}\;\mathrm{Value}=6^{2-2\;\mathrm{sensitivity}}$$

In this study, we used the PELT to detect change points. Because change point detection based on the slope (step 2) requires conservative control sensitivity, we defined the WUE sensitivity as 0.3, with which only one change point is detected at most.

#### 2.3.4. Hurst Index Analysis

The Hurst index (H) is used in prediction studies across demographics, economics, and climate change. R/S analysis helps to analyze the historical memory and fractal characteristics of time series. R/S stands for rescaled range analysis, which assesses a time series’ volatility and long-term dependence by comparing the range to the standard deviation. The Hurst phenomenon indicates long-term dependence and self-similarity, where past behavior influences future trends. A Hurst exponent greater than 0.5 suggests trend continuation, less than 0.5 indicates trend reversal, and equal to 0.5 means no memory effect. The Hurst phenomenon is commonly used in the analysis of climate change, financial markets, and other fields [70]. For example, if the time series is (WUE(t)), t = 1, 2, 3 …, n, and τ ≥ 1, then the average value of the sequence is as follows:(7)$$\bar{{WUE}_{\left(t\right)}}=\frac{1}{\tau }\sum_{t=1}^{\tau }{WUE}_{\left(\tau \right)},\tau =1,2,\ldots ,n$$(8)$$X_{\left(t,\tau \right)}=\sum_{t=1}^{t}\left({WUE}_{\left(t\right)}-\bar{{WUE}_{\left(t\right)}}\right),1\leq t\leq \tau$$(9)$$R_{\left(\tau \right)}={max}_{1\leq t\leq \tau }X_{\left(t,\tau \right)}-{min}_{1\leq t\leq \tau }X_{\left(t,\tau \right)},\tau =1,2,\ldots ,n$$(10)$$S_{\left(\tau \right)}={\left[\frac{1}{\tau }\sum_{t=1}^{\tau }{\left({WUE}_{\left(t\right)}-\bar{{WUE}_{\left(t\right)}}\right)}^{2}\right]}^{\frac{1}{2}},\tau =1,2,\ldots ,n$$

To confirm the presence of the Hurst phenomenon, the condition r must satisfy the R/S criteria. The Hurst index (H) was calculated using least-squares regression: log(R/S) n = a + H × log(n). The H values indicate whether the WUE time series is continuous or random. When H is between 0 and 0.5, the trend is unsustainable, meaning that future trends reverse past ones. When H is between 0.5 and 1, the trend is persistent, with future trends aligning with past ones. An H value of 0.5 suggests a random WUE trend.

In our study, the annual WUE served as the input for the model. We applied Theil–Sen median trend analysis and the Mann–Kendall test to quantify significant trends in water use efficiency. The resulting Hurst index was incorporated into the analysis of significant WUE differences, and the spatial differentiation of future WUE trends was predicted.

#### 2.3.5. Geographical Detector Model

The geographical detector model analyzes the relationships between factors from a spatial heterogeneity perspective. In this study, the WUE of the Mongolian Plateau (dependent variable) is influenced by meteorological and vegetation factors (independent variables). The model includes four detectors: factor, interaction, risk, and ecological [71]. It quantifies the explanatory power of different factors, helping to identify key influences and spatial patterns. We used factor and interaction detectors to measure how the independent variables explained the WUE and analyze their interactions, with the degree of interpretation expressed by the metric q [72], which is expressed as follows:(11)$$q=1-\frac{\sum_{h=1}^{K}N_{h}\sigma_{h}^{2}}{N\sigma^{2}}=1-\frac{SSW}{SST},\;SSW=\sum_{h=1}^{K}N_{h}\sigma_{h}^{2},\;SST=N\sigma^{2}$$
where q is the degree of interpretation of factors X to Y, and the range of q is [0, 1]. h = 1, 2, 3 …; K represents the grade of the independent or dependent variable; $\sigma^{2}$ and $\sigma_{h}^{2}$ are the variance of Y in the whole region and layer h, respectively; and Nh and N represent the number of layers h and the number of cells, respectively.

#### 2.3.6. Autoregressive Model

The autoregressive (AR) model identifies temporal dependencies in time-series data, being crucial in predicting trends and seasonal variations [73,74]. To analyze the short-term WUE stability, we removed long-term trends and seasonal components by subtracting the 37-year monthly average, calculating anomalies in WUE_Season_, NDVI, and meteorological factors during the growing season, and detrending abnormal time series. Finally, the seasonality of the non-trend abnormal time series was removed through z-score standardization [75]. An autoregressive model was established by referring to the method of Xie et al. [76], and NDVIA, PREA, TEMA, SWA, and DSIA were introduced. Please refer to the attachment for the formula used. Finally, applying the autoregressive model to all pixels in the Mongolian Plateau, we obtained the spatial distribution of the two metrics representing the resilience and resistance of the ecosystem WUE and removed the predictive variables with insignificant coefficients (*p* > 0.05).

#### 2.3.7. Residual Analysis Method

Residual analysis was used to calculate the impact of human factors on the WUE. Currently, this method is widely used to determine the impacts of climate change and human factors. Correlations among the WUE, precipitation, and temperature were calculated, and a regression relationship was established. Without considering the influence of other non-decisive factors, the difference between the time-series WUE values and the fitted and predicted WUE values based on the regression relationship was called WUEresidual. The calculation formula was as follows:(12)$${WUE}^{\prime }=aPRE+bTEM+cSW+dDSI+e$$(13)$${WUE}_{Residual}=WUE-{WUE}^{\prime }$$
where ${WUE}^{\prime }$ is the WUE predicted by the regression model, and a, b, c, d, and e are the model parameters. If the residual change is significant, it is considered that the WUE change is caused by non-meteorological factors; if the change in ${WUE}_{Residual}$ is not significant, it is considered that meteorological factors are the dominant causes of the changes in WUE in the region.

## 3. Results

### 3.1. Temporal and Spatial Trend Analysis of NPP, ET, and WUE

#### 3.1.1. Temporal Differentiation of WUE in the Mongolian Plateau

We studied the annual average WUE for different vegetation types in the Mongolian Plateau over 37 years and analyzed the linear trends (Figure 2). The results showed that the annual average WUE of forests was higher than that of grasslands, and the WUE for other vegetation types exhibited a decreasing trend, except for farmland (*p* < 0.05) and desert grasslands (*p* > 0.05). The annual rates of change in WUE in meadow, typical, and desert steppes; broadleaf forests; agricultural land; and coniferous forests were −0.005 gC/mm·m^2^·year, −0.002 gC/mm·m^2^·year, 0.0003 gC/mm·m^2^·year, −0.004 gC/mm·m^2^·year, 0.002 gC/mm·m^2^·year, and −0.003 gC/mm·m^2^·year, respectively.

#### 3.1.2. Spatial Distribution of WUE Dynamic Trends

Figure 3 shows the high NPP values on the Mongolian Plateau were mainly found in Northern and Northeastern Inner Mongolia. From 1982 to 2018, 13.03% of the pixels showed a significant decrease in NPP, mainly in Central and Northeastern Mongolia, while 17.41% showed significant growth, primarily in the sandy and agricultural areas of Northern and South–Central Inner Mongolia. High ET values were prevalent in Northern Mongolia and most of Inner Mongolia, with 47.17% showing significant growth, especially in the north, northeast, central, and southern regions. The WUE decreased significantly by 27.02%, mainly in Northern and Northeastern Inner Mongolia, while 10.26% of the pixels showed significant growth in the central and southern regions. The WUE decrease corresponded with a significant ET increase and a reduction in NPP. In areas with increased WUE, both NPP and ET grew significantly, but the NPP trend was stronger.

The annual WUE coefficient of variation showed the highest variability in the desert grassland belt, with the forest area being the most stable. From 1982 to 2018, 17.07% of the regions showed a significant decline in the DSI, mainly in the coniferous forests of Central and Western Mongolia, while 10.27% showed significant growth. The regions with significant DSI growth overlapped with areas of increased NPP and WUE, indicating that humid regions contributed to improvements in NPP and WUE.

Overall, from 1982 to 2018, the NPP decreased in Northern and Northeastern Mongolia, while the WUE increased in the central and southern regions. The highest WUE variability occurred in desert grasslands. Based on the BFAST model results in Figure 3d–h, combined with the SEN MK trend analysis, we analyzed the mutation trends of different vegetation types in the Mongolian Plateau. The detailed results are shown in Appendix A.

#### 3.1.3. Breakpoint Detection of WUE Time Series in the Growing Season

Figure 4 shows the trend analysis and breakpoint detection of WUE_Season_, NPP, ET, and 8 d WUE_Season_ during the growing season (April to October) on the Mongolian Plateau using the BFAST model. WUE_Season_ showed significant growth in 1997–1998, 2007–2009, and 2009–2018, especially in Eastern Mongolia and Central Inner Mongolia. The NPP_Season_ changes followed the WUE_Season_ trends, while ET_Season_ decreased during WUE_Season_ growth, suggesting that increased WUE_Season_ was linked to higher NPP_Season_ and lower ET, likely due to human factors.

#### 3.1.4. Detection of Change Points in Trends in Annual WUE

Breakpoints were detected only in typical steppes and broadleaf forests (Figure 4a–h). Steppes showed a sharp increase in 1992 and a decrease in 1999, with no breakpoints in NPP_Season_ and WUE_Season_, possibly due to human disturbances. Broadleaf forests showed growth from 1999 to 2017, with no significant decrease between 2017 and 2018. The first WUE_Season_ surge occurred from 1997 to 1998, with a rate of 0.135 gC/m^2^·mm·year, primarily in Eastern Mongolia and Central Inner Mongolia. A second surge happened from 2007 to 2009, at a rate of 0.121 gC/m^2^·mm·year, observed mainly in Eastern and Southern Mongolia. From 2009 to 2018, WUE_Season_ showed significant growth (0.009 gC/m^2^·mm·year). Both sharp increases in WUE_Season_ coincided with increased NPP_Season_ and decreased ET_Season_. No short-term WUE_Season_ changes were observed in the annual trend analysis, suggesting that human factors may have caused these short-term mutations. Overall, the WUE_Season_ on the Mongolian Plateau showed significant increases in 1997–1998, 2007–2009, and 2009–2018, particularly in Eastern Mongolia and Central Inner Mongolia, linked to higher NPP_Season_ and lower ET.

#### 3.1.5. WUE Trends

We determined the change points of the WUE over 37 years (Figure 5a–c). Figure 5a shows that change points occurred in most years, with 20.45% of the pixels showing changes between 1984 and 1988. Other significant periods were 2014–2018 (18.04%) and 1999–2013 (17.59%). Change points were also detected in 1994–1998 (15.28%), 2004–2008 (11.98%), and 2009–2013 (11.03%). A spatiotemporal variation analysis based on these breakpoints revealed that, between 1982 and 1997, 20.28% of areas had a highly significant increase in WUESeason. Between 1998 and 2007, 15.38% saw significant growth, while, between 2009 and 2018, 8.17% experienced a significant decline.

Combining the WUE trends from the Theil–Sen analysis with the Hurst index, we classified future WUE trends into six types: “persisting decrease”, “anti-persisting decrease”, “persisting increase”, “anti-persisting increase”, “steady”, and “unidentified”. The results showed that 25.07% of the pixels were in the “anti-persisting decrease” category, mainly in the forested areas of Northern and Northeastern Inner Mongolia and meadow grasslands in Northeastern Mongolia. “Persisting decrease” areas accounted for 1.96%, indicating potential future increases in WUE in forested regions. “Anti-persisting increase” regions constituted 9.96%, primarily in Southern Inner Mongolia, while “persisting increase” areas constituted only 0.87%, suggesting a potential overall decline in WUE in the Mongolian Plateau.

### 3.2. Potential Factors Affecting WUE Changes

#### 3.2.1. Correlations Between WUE and Meteorological Factors (PRE, TEM, SW, and DSI) on the Mongolian Plateau

We estimated the Hurst index of the annual WUE data for the Mongolian Plateau (Figure 6a). Areas with a Hurst index less than 0.5, greater than 0.5, and equal to 0.5 accounted for 92.78%, 6.22%, and 0%, respectively. This suggests that most areas on the Mongolian Plateau may experience a reversal in their WUE trends in the future. Before the change point, 42.5% of pixels showed a decline and 57.4% showed an increase. After the change point, 34.95% showed a decline and 65.05% showed an increase, indicating that the WUE reversed from a declining trend to an increasing one in 7.65% of the pixels, mainly in broadleaved forests in Northeastern and Central Inner Mongolia (Figure 6b).

As shown in Figure 7a, 88.47% of the pixels of PRE and WUE showed a positive correlation, 36.22% of which were significantly correlated (*p* < 0.05). These pixels were mainly distributed in Eastern and Northeastern Inner Mongolia. This indicated that an increase in precipitation contributed to an increase in WUE. In total, 77.22% of the pixels of PRE and WUE showed a negative correlation, and only 9.58% of these correlations were significant (*p* < 0.05). These pixels were mainly distributed in the broadleaf forest areas of Eastern and Northeastern Inner Mongolia. TEM and WUE were generally negatively correlated; areas with this negative correlation were mainly distributed in the middle and east of Mongolia and the northeast and middle of Inner Mongolia, but their significance was not high (Figure 7b).

Pixels were distributed in the meadow steppe belt in Eastern Mongolia and in the central and southern areas of Inner Mongolia; however, only 4.62% of these pixels were significant (*p* < 0.05). A total of 85.47% of the pixels of SW and WUE showed a positive correlation, of which 35.28% were significant (*p* < 0.05). These pixels were mainly distributed in the meadow steppe belt in Eastern Mongolia and Eastern and Southeastern Inner Mongolia (Figure 7c). This part of the region overlaps with areas with a significant positive correlation between PRE and WUE and with areas with a negative correlation between TEM and WUE. This indicates that water is an important factor driving the change in WUE in this area, and that increases in temperature and water consumption will lead to a decrease in WUE. Therefore, in the arid and semiarid areas of the Mongolian Plateau, water is very important for changes in WUE. Water loss led to a decrease in NPP, and a change in temperature led to an increase in ET, which in turn led to a decrease in WUE.

On the Mongolian Plateau, 93.57% of the pixels of DSI and WUE showed a negative correlation; 53.03% of these correlations were significant (*p* < 0.05) and they were mainly distributed in the central and eastern parts of Mongolia and most areas of Inner Mongolia, except for the northeast (Figure 7d). This shows that drought caused by water loss is the main factor in reducing the WUE and that moisture is helpful in increasing the WUE. However, there was a negative correlation between DSI and WUE in the forested areas of Northeastern Inner Mongolia, which may have been caused by significant growth in ET in this area. Phenological vegetative parameters are important indicators of vegetative growth. At the same time, the change in the growth cycle of the vegetation itself is fed back to the NPP and ET accordingly; therefore, the change in phenological parameters may also affect the change in WUE. Overall, PRE and SW were positively correlated with WUE in most areas of the Mongolian Plateau, while TEM and DSI were negatively correlated with WUE, particularly in the central and eastern regions. Water loss and increased temperatures led to decreased WUE, especially in arid and semiarid areas. SOS showed a positive correlation with WUE, NPP, and ET, with delayed SOS improving WUE and NPP, while EOS had a minimal effect on WUE.

#### 3.2.2. Geographical Detection Model for WUE Drivers

The factor detector results showed that the highest degree of interpretation of NPP for WUE was 0.9066 (Figure 8a). The factors with q-values greater than 0.5 included NDVI (q = 0.7266), SW (q = 0.6009), and PRE (q = 0.5281), while TEM and ET had values below 0.5. SOS, EOS, and DSI had the lowest interpretation degrees, with q-values of less than 0.1. WUE is mainly influenced by PRE, TEM, SW, DSI, EOS, SOS, and NDVI, with NDVI and PRE showing collinear enhancements in WUE, as well as NDVI and TEM. PRE and SOS, as well as SW and EOS, have non-collinear enhancement effects on WUE. Thus, vegetation has a greater effect on WUE than water. The interaction detector results revealed that, apart from SOS and EOS, all other factors had nonlinear enhancement effects, and their interactions showed a linear enhancement effect on WUE.

#### 3.2.3. Spatial Distribution of WUE Anomaly Resilience and Its Resistance to Vegetation and Meteorological Factor Anomalies

It can be seen from the results in Figure 9a that the overall resilience of WUEA in the Mongolian Plateau is strong. The resilience of forest areas in Northeastern Inner Mongolia, especially broadleaved forest areas, as well as sandy land and artificial vegetation in the middle of Inner Mongolia, is weak, and the resilience of grassland ecosystems is strong as a whole. In addition, the forest NDVIA had a significant positive contribution to WUE, whereas the grassland ecosystem NDVIA had a significant negative contribution. WUE had weak resistance to NDVIA, mainly in humid forest areas and arid desert grassland zones.

WUE had strong resistance to PREA overall, and only the desert steppes around the Gobi Desert and some agricultural land in the middle of Inner Mongolia had significant resistance (Figure 9c). WUE showed weak resistance to TEMA (Figure 9d). WUE was significantly positively correlated with TEMA in Central and Western Mongolia and Northeastern Inner Mongolia. A significant negative correlation was observed between Southern Mongolia and Central and Southern Inner Mongolia. The results show that TEMA has a strong effect on WUE and that it is not easy to recover from this. At the same time, the temperature anomalies in the cold, high-latitude areas and the warm, low-latitude areas show the opposite trends in the interference of WUE. WUE also had strong resistance to SWA as a whole, especially in grassland ecosystems, but its significance was not high. However, its resistance to SWA disturbances in forest ecosystems in Mongolia and Northeastern Inner Mongolia was relatively weak (Figure 9e). WUE was weakly resistant to DSIA. The DSIA of grassland ecosystems made a significant positive contribution to WUE, whereas that of forest ecosystems made a significant negative contribution to WUE (Figure 9f). Overall, the WUE in the Mongolian Plateau had strong resistance to water anomalies, but weak resistance to temperature, drought, and vegetation greenness anomalies.

Overall, the WUE on the Mongolian Plateau shows strong resilience to water anomalies but weak resistance to temperature, drought, and vegetation greenness anomalies, with forest ecosystems being more sensitive to temperature and drought disturbances.

#### 3.2.4. The Impact of Human Activities on WUE Based on Residual Analysis

Human activities affect WUE when excluding meteorological factors. The results showed that WUE_Residual_ was positive in forest areas, especially coniferous forests at high latitudes, suggesting that human activities may increase the forest WUE. In contrast, WUE_Residual_ was negative in grasslands, especially in Western Mongolia and Central–Southern Inner Mongolia, indicating a reduction in the grassland WUE (Figure 10a). Over 37 years, 9.91% of the WUE_Residual_ pixels in Northern Mongolia’s coniferous forests and eastern grasslands showed a significant reduction, while 9.26% in Central–Southern Inner Mongolia showed a significant increase. These changes overlapped with areas of significant annual WUE changes, suggesting that human activities contributed to WUE variations (Figure 10b).

## 4. Discussion

In this study, we analyzed the linear trends in WUE and the trends and alterations in the WUE_Season_. We found that the trend in WUE over the whole year was significantly reduced; however, the trend in the WUE_Season_ was significantly enhanced because NPP weakly contributed to WUE in the non-growing season. Simultaneously, the ET during the non-growing season decreased the overall WUE for the entire year. Spatial differentiation is important in analyzing the effects of ET and NPP on WUE. For example, if an area with significant ET growth also exhibits significant NPP growth, the change in WUE depends on the proportional growth rates of NPP and ET. However, sometimes, the NPP decreases significantly in areas with significant growth in ET, which leads to a significant reduction in WUE; this has already occurred in the meadow grasslands in Northeastern Mongolia (Figure 4b,d,f). The results of the breakpoint analysis showed that the spatial distribution of the sudden increasing trend in WUE_Season_ was not the same over the past 37 years. The two alterations that occurred between 1997 and 2009 were mainly distributed in Eastern Mongolia and Central Inner Mongolia. During this period, the WUE_Season_ increased significantly, and these regions overlapped with the WUE, PRE, and especially the DSI regions, which were significantly and positively related. This shows that the steep increase in WUE_Season_ before 2009 may have been related to changes in water content. An extremely significant increase from 2010 to 2018 was observed in the central and southern regions of Inner Mongolia, where more artificial vegetation was distributed; these regions overlapped with regions where WUE_Resideal_ significantly increased (Figure 10b). Human activities inevitably interfere with the growth of vegetation [64,77] and even indirectly interfere with ET values [66] owing to changes in vegetation transpiration, which has been confirmed in previous studies. The results show that human activity may lead to an increase in WUE in forests and a decrease in WUE in grasslands. Moreover, as shown by the results of the annual WUE trend analysis, the most significant changes were observed in agricultural land (with more artificial vegetation) and desert vegetation in Central and Southern Inner Mongolia [30]. This shows that human activities have greatly contributed to the sharp increase in WUE_Season_ after 2010. The results of the trend change point detection showed that, except from 1989 to 1993, the trend in the WUE of the Mongolian Plateau changed in most years. In addition, the broadleaf forest areas in Northeastern and Eastern Inner Mongolia reversed from an original decreasing trend to an increasing trend. The analysis of WUE trends at different spatiotemporal scales will lead to different information.

Among the meteorological factors, changes in water significantly affected WUE, whereas the effect of the change in temperature was not significant. This is consistent with the results of previous studies [30]. An increase in environmental humidity also significantly improves WUE, mainly in grassland ecosystems. The DSI had the most significant positive contribution to WUE, indicating that drought greatly affects WUE. In addition, an increase in the DSI will have an insignificant effect on wetter forest areas. In contrast to the Pearson correlation analysis results for the interannual WUE and meteorological factors, WUE had strong resistance to abnormal water disturbances but weak resistance to anomalous temperature disturbances. In addition, the positive and negative contributions of TEM to WUE were strongly influenced by the latitude. Forest areas at high latitudes were usually positively correlated, whereas grasslands at low latitudes were negatively correlated. This shows that the temperature has no significant impact on WUE under general conditions; however, once the temperature changes at the lower end, its impact on the WUE and water content becomes significant. In warm low-latitude areas, high temperatures lead to water evaporation, which not only causes water stress and affects vegetation development but also increases ET, thus placing double pressure on WUE. However, forest areas at high latitudes are cold and humid, and the ET is usually high, whereas low temperatures inhibit the photosynthesis of vegetation, thus reducing the NPP. Water anomalies had the same effect (either increasing or decreasing) on the NPP and ET, which may mean that water has a stable influence on WUE; thus, WUE has strong resistance to water disturbances. The DSI and NDVI had a spatial distribution that was similar to that of WUE, especially regarding the difference between forests and grasslands; however, the opposite trend was observed. The NDVI in forest areas was positively correlated with WUE, which indicates that the poorer the forest vegetation growth, the more WUE will be promoted, while drought anomalies in forest areas will increase the WUE (Figure 9b,f). This trend is reversed in grasslands, which shows that the abnormal synergism of drought and vegetation has significantly different effects on WUE in areas with different vegetation types.

From the perspective of spatial differentiation, the degree of interpretation of vegetation change regarding WUE was higher than that for meteorological factors (Figure 8a), which is consistent with previous findings [78]. In addition, delaying the rejuvenation period will eventually affect WUE to some extent. The earlier the SOS in spring, the earlier the plants begin to photosynthesize, leading to a rapid and sustained increase in GPP. However, the changes in ET during spring are unclear. Earlier leaf expansion or a larger leaf area in spring can increase plant transpiration; however, evaporation decreases because of the reduced surface temperature and solar radiation. This resulted in a smaller increase in ET than in GPP, ultimately leading to an improvement in WUE in spring. Therefore, in summary, the effect of ET, which was more significantly affected by the vegetation phenology, on WUE was compromised by NPP. Moreover, the cooperation of phenological parameters and meteorological factors had a nonlinear enhancing effect on WUE.

The WUE of terrestrial ecosystems reflects the interactions between vegetation productivity and water availability. Because climate change leads to increased precipitation variability, precipitation infiltration directly affects the replenishment of soil moisture. These factors significantly influence ecosystem evapotranspiration processes, thereby impacting vegetation growth and the water–carbon cycle and ultimately resulting in changes in WUE across different vegetation types. Future climate warming will further exacerbate the imbalance between the water supply and demand in terrestrial ecosystems. Therefore, understanding the changes in WUE under different future scenarios is crucial to ensure the sustainable use of water resources. Future studies should consider precipitation, soil moisture, and evapotranspiration under 1.5 °C and 2.0 °C warming scenarios as key indicators for the assessment of ecosystem WUE, thereby improving our understanding of the impacts of climate change on the ecosystems of arid and semiarid regions on the Mongolian Plateau.

## 5. Conclusions

In this study, we used a variety of statistical methods to analyze the temporal and spatial dynamic changes in NPP, ET, WUE, and DSI on the Mongolian Plateau from 1982 to 2018 and evaluated the response of WUE to changes in meteorological factors, vegetation, and human intervention.

The following are the main conclusions of our work.

The annual average NPP and WUE were 232.75 gC/m^2^·year and 0.592 gC/mm·m^2^·year, respectively. Over 37 years, the annual WUE decreased significantly, while the annual ET increased. Significant WUE decreases were observed in Central and Eastern Mongolia and in broadleaved forests in Northeastern Inner Mongolia. Significant increases were found in Central and Southern Inner Mongolia. Two WUE_Season_ surges were detected in 1997–1998 and 2007–2009. Some broadleaf forests in Inner Mongolia reversed their decreasing WUE trends in winter. The overall trend suggests that the WUE may shift from decreasing to increasing in the future. The Mongolian Plateau showed strong WUE resilience, except for artificial vegetation areas in Central Inner Mongolia.WUE was significantly affected by precipitation and soil moisture, showing resistance to anomalous water disturbances. It had a weak correlation with the temperature and limited resistance to temperature disturbances. WUE was positively correlated with the DSI, although its resistance to DSI anomalies was weak.Vegetation change had a stronger impact on WUE than meteorological factors. WUE showed weak resistance to anomalous NDVI disturbances. Delayed rejuvenation positively influenced WUE. Interactions between phenological parameters and meteorological factors enhanced WUE nonlinearly. Human activities significantly contributed to WUE increases in Eastern, Central, and Southern Inner Mongolia.

## Figures and Tables

**Figure 1 sensors-25-02214-f001:**
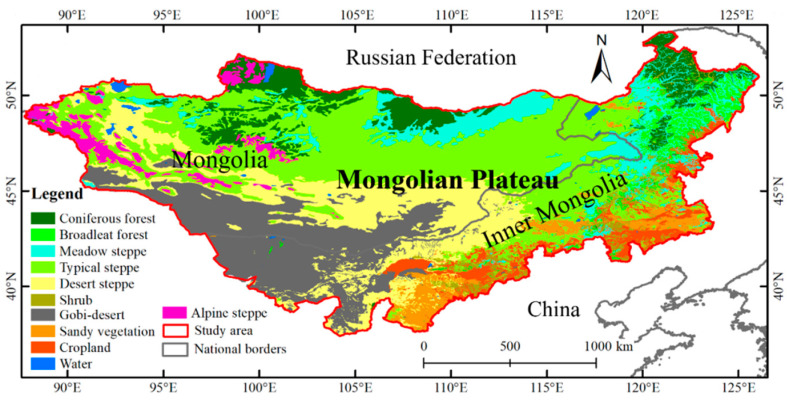
The land type distribution map of the Mongolian Plateau.

**Figure 2 sensors-25-02214-f002:**
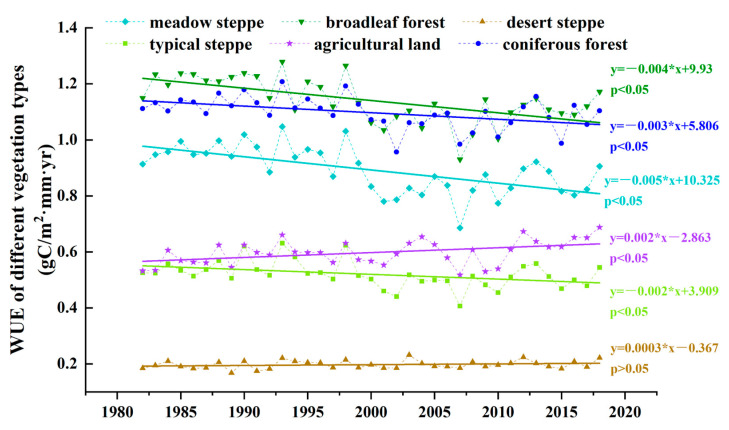
Annual variation in water use efficiency for different vegetation types in the Mongolian Plateau.

**Figure 3 sensors-25-02214-f003:**
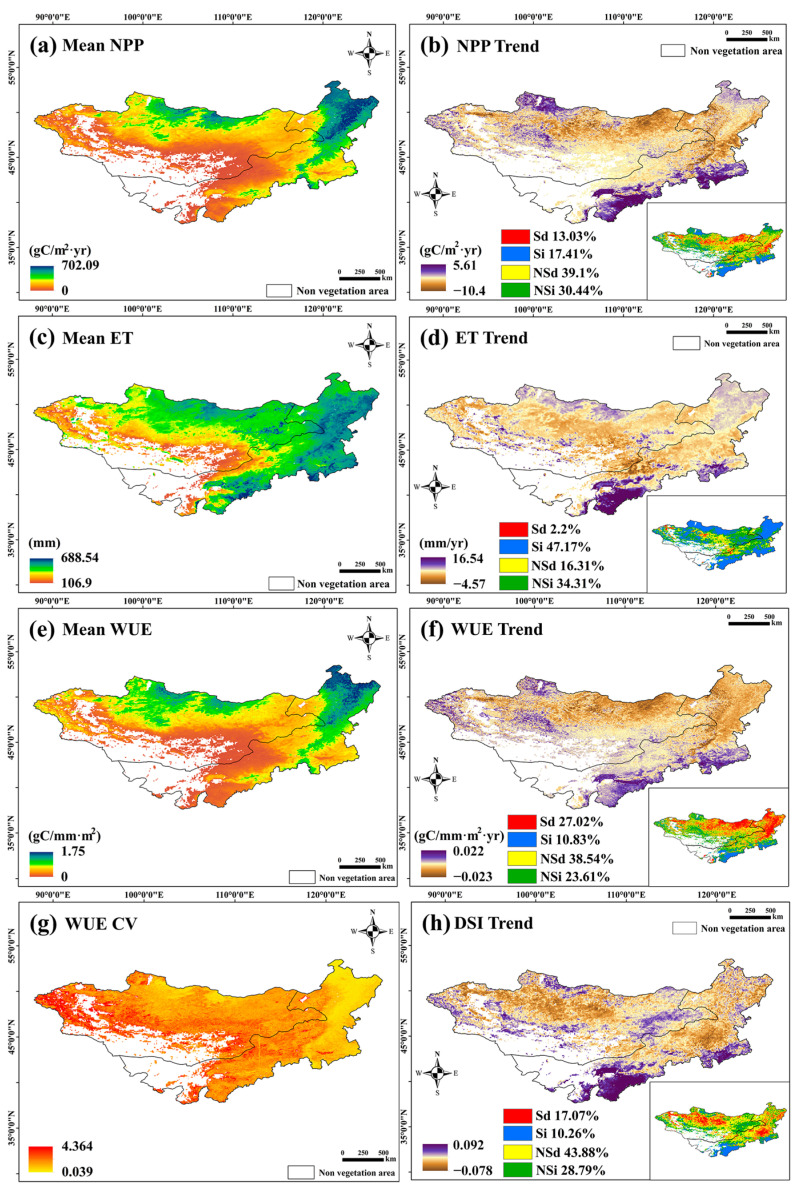
Spatial trend analysis of WUE on the Mongolian Plateau over the past 37 years. (**a**) shows the annual average values of WUE; (**b**) shows the spatial distribution of the coefficient of variation in WUE; (**c**) shows the average ET over the years; (**d**) shows ET trend; (**e**) shows the average WUE over the years; (**f**) shows WUE trend; (**g**) shows the coefficient of variation of WUE; (**h**) shows DSI trend Note: Sd indicates a significant decrease, Si indicates a significant increase, NSd indicates a nonsignificant decrease, and NSi indicates a nonsignificant increase.

**Figure 4 sensors-25-02214-f004:**
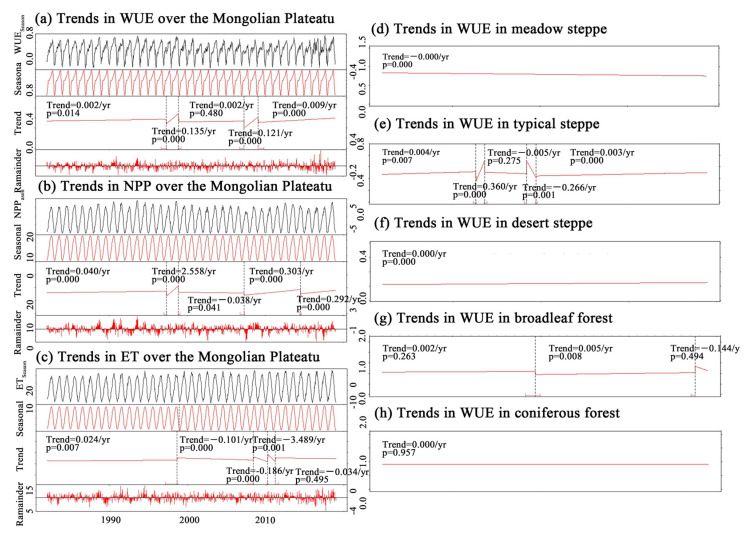
The trend analysis and breakpoint detection over 8 d: (**a**) WUE_Season_, (**b**) NPP_Season_, and (**c**) ET_Season_ in the Mongolian Plateau; (**d**–**h**) show the trend analysis and breakpoints of the 8 d WUE_Season_ for meadow steppes, typical steppes, desert steppes, broadleaf forests, and coniferous forests, respectively.

**Figure 5 sensors-25-02214-f005:**
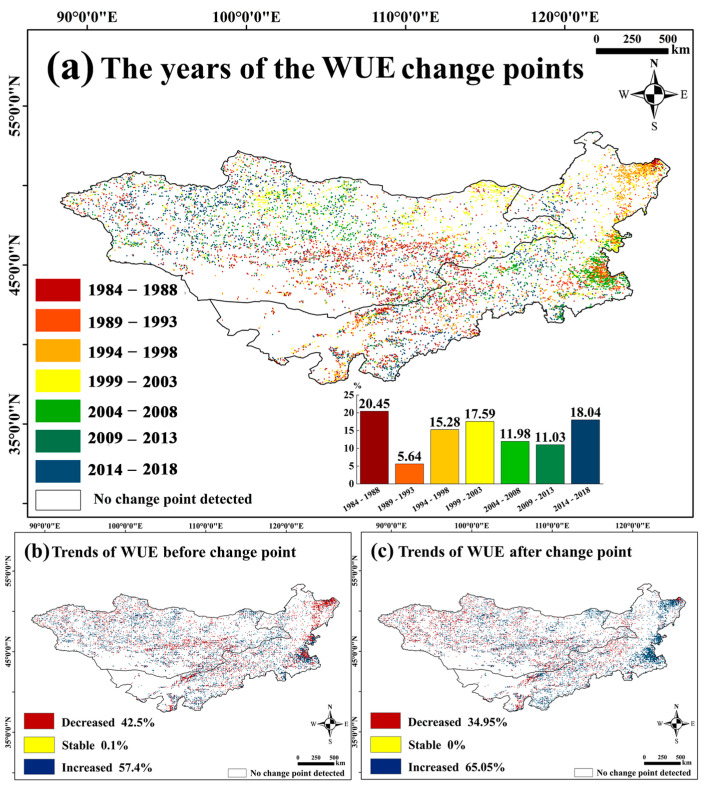
(**a**) Change points of WUE in the Mongolian Plateau; (**b**) trends before and (**c**) trends after the change points, respectively.

**Figure 6 sensors-25-02214-f006:**
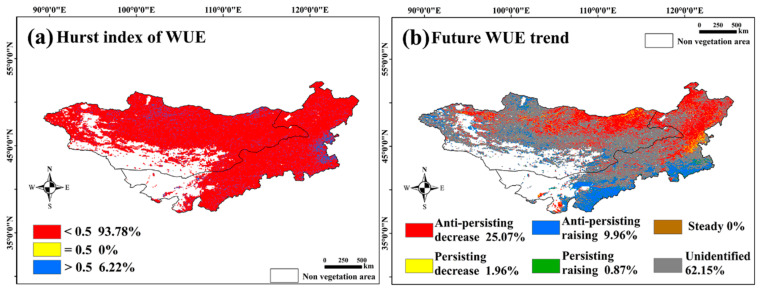
Hurst index: (**a**) WUE and (**b**) future WUE trends in the Mongolian Plateau from 1982 to 2018.

**Figure 7 sensors-25-02214-f007:**
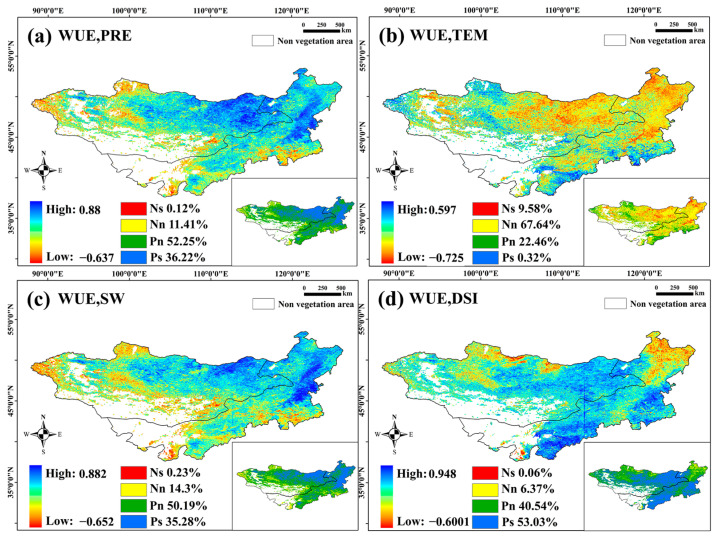
Partial correlation analysis between WUE and meteorological factors, where (**a**–**d**) show the partial correlation coefficients and significance analysis of PRE, TEM, SW, and DSI with WUE, respectively. Note: Ns indicates a significant negative correlation, Nn indicates a nonsignificant negative correlation, Pn indicates a nonsignificant positive correlation, and Ps indicates a significant positive correlation.

**Figure 8 sensors-25-02214-f008:**
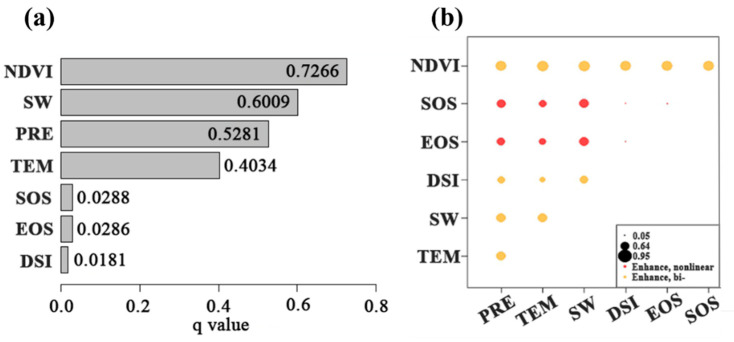
Geographical detection model: (**a**) calculation results of the factor detector and (**b**) factor interaction detector results between each driving factor and the annual WUE.

**Figure 9 sensors-25-02214-f009:**
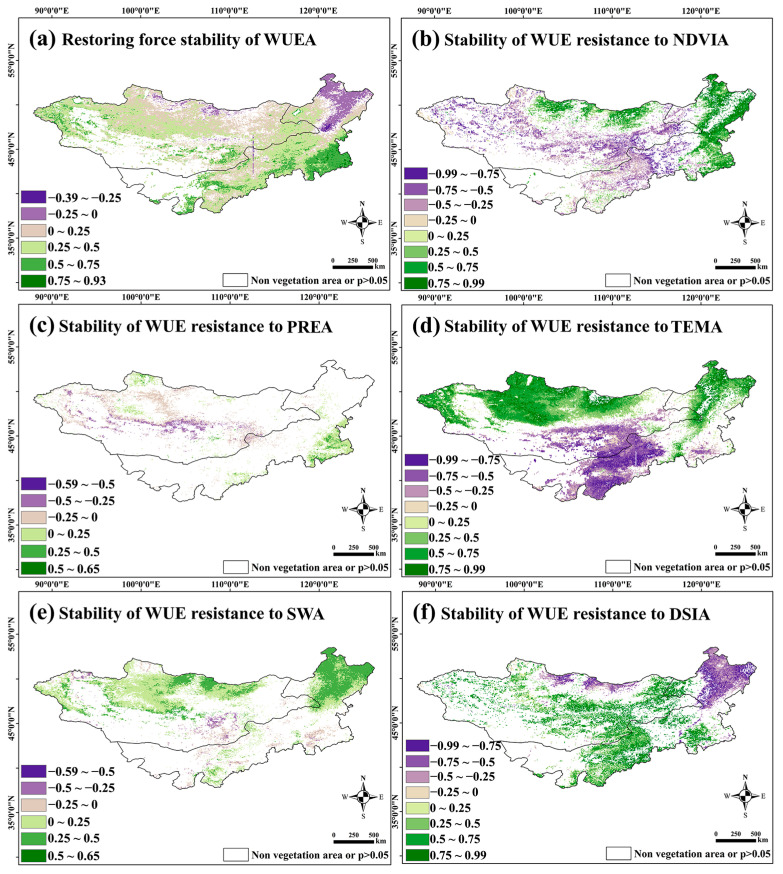
(**a**) shows the resilience index of WUE; (**b**–**f**) are the resistance indices of WUE regarding PREA, TEMA, SWA, and DSIA disturbances, respectively.

**Figure 10 sensors-25-02214-f010:**
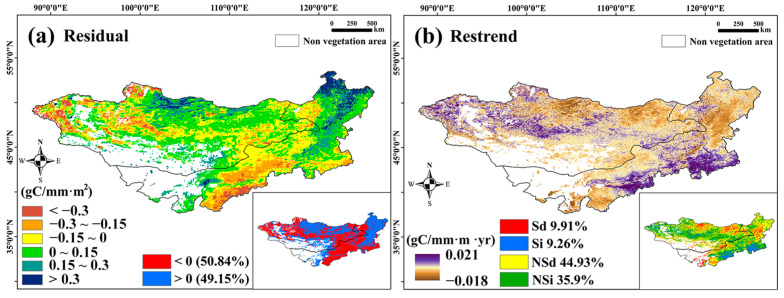
(**a**) shows the 37-year average value of WUE_Residual_ in the Mongolian Plateau; (**b**) shows the spatial distribution of the WUE_Residual_ trend from 1982 to 2018.

## Data Availability

Data available on demand.

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
