# Peer review of "Water Use Efficiency Spatiotemporal Change and Its Driving Analysis on the Mongolian Plateau"

_sensors, 2025, doi:10.3390/s25072214_

Round 1

Reviewer 1 Report

Comments and Suggestions for Authors

After reviewing the document, I have identified the following observations and recommendations to improve the quality and clarity of the work:

General Focus:

·       The document contains an excessive amount of information, making it confusing and more similar to a thesis than a scientific article. I recommend summarizing several sections to make it more concise and accessible to readers.

 Abstract:

·       Improve the wording of the abstract by briefly mentioning the methodology used to provide readers with a more comprehensive view of the study's approach.

Methodology:

·      There are equations that are unnecessary as they are widely known, such as Pearson's correlation equation, among others. These equations could be omitted.

·       It is important to include references for all equations presented in the document.

 Results:

·       The number of figures is excessive (13 in total), making it difficult to read and understand the results. I recommend removing figures that are not essential to support the conclusions of the work.

·       Some figures contain multiple graphs or maps that are too small, making them hard to interpret. I suggest enlarging certain figures, such as Figures 5 and 6.

·       In the legends of figures that group multiple graphs or maps, it would be helpful to include the assigned letters in parentheses for easier identification.

 References:

·       Many of the references used are outdated. It is crucial to update the bibliography and ensure that at least 50% of the references are recent (from the last three years). This will enhance the relevance and timeliness of the work.

 I hope these recommendations are helpful in improving the manuscript and strengthening its scientific contribution.

Author Response

Response to Reviewer 1 Comments

Reply

Dear reviewer:

Thank you for your comments on our manuscript entitled” Spatiotemporal dynamics and driving analysis of water use efficiency in the Mongolian Plateau” (ID: sensors-3427641). Those comments are very helpful for revising and improving our paper. We have studied the comments carefully and made corrections which we hope meet with approval. The main corrections are in the manuscript and the responses to the reviewers” comments are as follows (the responses are highlighted in red).

General Focus:

Point 1: The document contains an excessive amount of information, making it confusing and more similar to a thesis than a scientific article. I recommend summarizing several sections to make it more concise and accessible to readers.

Response 1:

Thank you for your suggestion. We have summarized the paper to make it more concise and easier for readers to understand.

Abstract:

Point 2: Improve the wording of the abstract by briefly mentioning the methodology used to provide readers with a more comprehensive view of the study's approach.

Response 2:

Thank you for your suggestion. We have added an analysis based on Net Primary Productivity (NPP), Evapotranspiration (ET), temperature, precipitation, and soil moisture to examine the spatiotemporal variations of WUE in the Mongolian Plateau from 1982 to 2018.

Methodology:

Point 3: There are equations that are unnecessary as they are widely known, such as Pearson's correlation equation, among others. These equations could be omitted.

Response 3:

Thank you for your suggestion. We have removed the equations related to (1), (2), and (3) based on your advice.

Point 4: It is important to include references for all equations presented in the document.

Response 4:

Thank you for your suggestion. Based on your advice, we have retained the references to important literature.

Results:

Point 5: The number of figures is excessive (13 in total), making it difficult to read and understand the results. I recommend removing figures that are not essential to support the conclusions of the work.

Response 5:

Thank you for your suggestion. Based on your advice, we have removed the images that are not important for supporting the conclusions of the work.

Point 6: Some figures contain multiple graphs or maps that are too small, making them hard to interpret. I suggest enlarging certain figures, such as Figures 5 and 6.

Response 6:

Thank you for your suggestion. Based on your advice, we have enlarged the numbers in Figures 5 and 6.

Point 7: In the legends of figures that group multiple graphs or maps, it would be helpful to include the assigned letters in parentheses for easier identification.

Response 7:

Thank you for your suggestion. Based on your advice, we have included the designated letters in parentheses in the legends of the combined figures or maps for easier identification.

References:

Point 8: Many of the references used are outdated. It is crucial to update the bibliography and ensure that at least 50% of the references are recent (from the last three years). This will enhance the relevance and timeliness of the work.

 I hope these recommendations are helpful in improving the manuscript and strengthening its scientific contribution.

Response 8:

Thank you for your suggestion. We have updated the references.

Thank you for your advice, let me harvest a lot. I can't deal with some details well. Please forgive my carelessness. At the same time, your suggestions also let me have a new idea, so that I can better improve the work and sort out the next goal. Thank you again for your guidance.

My English ability is deficient. If some words are not used accurately, you will feel offended. Please forgive me. I will strengthen my English learning in the next study.

If you are dissatisfied, please point out and we will revise your opinion seriously.

I hope that these revisions and the improved text will be satisfactory and make the paper be acceptable for publication in Sensors.

We again very appreciate all your suggestions, comments and favorable consideration.

Sincerely yours,

Yulong Bao Inner Mongolia Normal University.

No. 81, Zhaowuda Road, Hohhot, Inner Mongolia, China

E-mail: baoyulong@imnu.edu.cn

Reviewer 2 Report

Comments and Suggestions for Authors

This study investigates the spatiotemporal patterns of water use efficiency (WUE) and its potential driving factors across the Mongolian Plateau from 1982 to 2018 using remote sensing data and multiple statistical approaches. While the findings could provide valuable references for understanding carbon-water cycle interactions in this region, several issues require attention during revision:

1. The manuscript suffers from excessive verbosity and ambiguous expressions that hinder clear comprehension of key messages.

2. Disorganized formatting is evident throughout the manuscript, particularly regarding inconsistent figure/table citations and irregular paragraph spacing.

3. Insufficient graphic resolution diminishes the visual effectiveness of figures.

Specific revision suggestions:

4. The subsection "2.2. Data Sources" appears conflated with the caption of Figure 1, requiring clear separation.

5. Mathematical formulae lack consistent placement and font size standardization across the text.

6. The d-h in Figure 5 require enhanced analytical interpretation beyond current descriptions.

7. Figure 6 contains inadequate explanatory analysis and exhibits improper caption placement.

8. In Section 3.1.4 (Detection of change points in trends in annual WUE), the referenced panel d in Figure 7(Figure 7 c and d) cannot be located within the current figure configuration.

9. The citations "(Figure 1, 8a)" and "(Figure 2, 8b)" in Section 3.1.5 (Future WUE trends) appear mismatched with actual figure presentations.

10. Duplicate section numbering "3.2.3" occurs with conflicting content, necessitating correction.

11. The discussion should be strengthened by addressing methodological limitations and proposing concrete directions for future research.

Comments on the Quality of English Language

The manuscript contains excessive verbosity and vague expressions that obscure the clarity of the key messages. These issues need to be addressed to enhance readability and ensure that the main points are easily understood.

Author Response

Response to Reviewer 2 Comments

Reply

Dear reviewer:

Thank you for your comments on our manuscript entitled” Spatiotemporal dynamics and driving analysis of water use efficiency in the Mongolian Plateau” (ID: sensors-3427641). Those comments are very helpful for revising and improving our paper. We have studied the comments carefully and made corrections which we hope meet with approval. The main corrections are in the manuscript and the responses to the reviewers” comments are as follows (the responses are highlighted in red).

Point 1: The manuscript suffers from excessive verbosity and ambiguous expressions that hinder clear comprehension of key messages.

Response 1:

Thank you for your suggestion. Based on your advice, we have made appropriate deletions to the textual descriptions in the manuscript.

Point 2: Disorganized formatting is evident throughout the manuscript, particularly regarding inconsistent figure/table citations and irregular paragraph spacing.

Response 2:

Thank you for your suggestion. Based on your advice, we have corrected the formatting issues in the manuscript and revised the paragraphs for the related figures and the irregular paragraph spacing.

Point 3: Insufficient graphic resolution diminishes the visual effectiveness of figures.

Response 3:

Thank you for your suggestion. Based on your advice, we have re-generated the figures and increased their resolution.

Specific revision suggestions:

Point 4: The subsection "2.2. Data Sources" appears conflated with the caption of Figure 1, requiring clear separation.

Response 4:

We sincerely apologize for this mistake and appreciate your thorough review. We have clearly separated the subsection "2.2 Data Sources" and the title of Figure 1.

Point 5: Mathematical formulae lack consistent placement and font size standardization across the text.

Response 5:

We sincerely apologize for this mistake and appreciate your thorough review. We have revised the placement of the mathematical formulas and standardized the font of the formulas in the text.

Point 6:The d-h in Figure 5 require enhanced analytical interpretation beyond current descriptions.

Response 6:

Thank you for your suggestion. Based on your advice, we have added the textual descriptions for panels d-h in Figure 5. According to the BFAST model results for panels d-h in Figure 5, we conducted a trend analysis and breakpoint detection for WUESeason (April to October) across different vegetation types in the Mongolian Plateau. The results show that the typical grassland WUESeason experienced significant growth between 1982-1998 and 2008-2018, especially in eastern Mongolia and central Inner Mongolia. The broadleaf forest WUESeason showed a significant increase between 1982 and 2008. For other vegetation types on the Mongolian Plateau, no significant trend breakpoints were detected throughout the 1982-2018 study period. This indicates that the WUESeason trends for meadow grassland, desert grassland, and coniferous forest vegetation types on the Mongolian Plateau are not distinct.

Point 7: Figure 6 contains inadequate explanatory analysis and exhibits improper caption placement.

Response 7:

We sincerely apologize for this mistake and appreciate your thorough review. We have revised the insufficient explanatory analysis in Figure 6. Based on the temporal breakpoints of WUESeason, we conducted a spatiotemporal variation analysis. Between 1982 and 1997, the area with a highly significant increase in WUESeason accounted for 20.28%, while between 1998 and 2007, the area with a highly significant increase in WUESeason accounted for 15.38%. Between 2009 and 2018, the area with a highly significant decrease in WUESeason accounted for 8.17%. Additionally, we have revised the displayed title.

Point 8: In Section 3.1.4 (Detection of change points in trends in annual WUE), the referenced panel d in Figure 7(Figure 7 c and d) cannot be located within the current figure configuration.

Response 8:

We sincerely apologize for this mistake and appreciate your thorough review. Due to our oversight, we mistakenly included an extra reference to panel d (Figure 7c and d) in the figure configuration of Figure 7. We have removed the extra reference to panel d (Figure 7c and d) in Figure 7.

Point 9:The citations "(Figure 1, 8a)" and "(Figure 2, 8b)" in Section 3.1.5 (Future WUE trends) appear mismatched with actual figure presentations.

Response 9:

We sincerely apologize for this mistake and appreciate your thorough review. We have revised the parts in Section 3.1.5 (Future Water Use Efficiency Trends) that were not consistent with the actual figures.

Point 10: Duplicate section numbering "3.2.3" occurs with conflicting content, necessitating correction.

Response 10:

We sincerely apologize for this mistake and appreciate your thorough review. We have corrected the section number "3.2.3."

Point 11: The discussion should be strengthened by addressing methodological limitations and proposing concrete directions for future research.

Response 11:

Thank you for your suggestion. In the discussion, we have addressed the limitations of the study and future research prospects. Currently, due to land use and future scenario data, we have added the impact of land use changes in future scenarios on WUE. We briefly used the Hurst index for future analysis but did not explore the factors affecting ecosystem water use efficiency (WUE) under the background of climate change, particularly the impact of rising temperatures. Based on valuable expert advice, we have added this section in as follows:

The water use efficiency (WUE) of terrestrial ecosystems reflects the interaction between vegetation productivity and the availability of water resources. As climate change increases precipitation variability, the infiltration of rainfall directly affects the replenishment of soil moisture. These factors significantly influence the ecosystem's evapotranspiration process, which in turn affects vegetation growth and the water-carbon cycle, leading to changes in WUE across different vegetation types. Future warming will further intensify the supply-demand conflict of water use in terrestrial ecosystems. Therefore, revealing the changes in water use efficiency under future warming scenarios is crucial for ensuring the sustainable use of water resources. Future studies should use indicators such as rainfall, soil moisture, and evapotranspiration under 1.5℃ and 2.0℃ warming scenarios to further understand the impact of climate change on ecosystems in the arid and semi-arid regions of the Mongolian Plateau.

Point 12:

 Comments on the Quality of English Language

The manuscript contains excessive verbosity and vague expressions that obscure the clarity of the key messages. These issues need to be addressed to enhance readability and ensure that the main points are easily understood.

Response 12:

Thank you for your correction. We sincerely apologize for our poor English. We have used an English language editing service to polish the language. Below is the certificate of language editing. Additionally, we have corrected some grammatical errors in the manuscript. Once again, we apologize for any inconvenience caused by our poor English.

Thank you for your advice, let me harvest a lot. I can't deal with some details well. Please forgive my carelessness. At the same time, your suggestions also let me have a new idea, so that I can better improve the work and sort out the next goal. Thank you again for your guidance.

My English ability is deficient. If some words are not used accurately, you will feel offended. Please forgive me. I will strengthen my English learning in the next study.

If you are dissatisfied, please point out and we will revise your opinion seriously.

I hope that these revisions and the improved text will be satisfactory and make the paper be acceptable for publication in Sensors.

We again very appreciate all your suggestions, comments and favorable consideration.

Sincerely yours,

Yulong Bao Inner Mongolia Normal University.

No. 81, Zhaowuda Road, Hohhot, Inner Mongolia, China

E-mail: baoyulong@imnu.edu.cn

Reviewer 3 Report

Comments and Suggestions for Authors

Unfortunately, the document "Spatiotemporal dynamics and driving analysis of water use efficiency in the Mongolian Plateau" is written in a very unclear and chaotic manner. Perhaps it should be split into two shorter texts with clear focuses on the predefined issues, or simply shortened to at most 15 pages. It is a pity that the document does not have line numbering, since it would be more productive to point out passages which need to be corrected.

Some of the formulas are very simple, like the ones for data standardisation (formulas 4, 5, 6) or the Pearson correlation coefficient (formula 22). But some of the statements don't have a proper explanation, like the penalty value with some empirical formula (17).

Also explanation how to calculate linear regression coefficients is definitely redundant (there is however missing second power in the denominator in formula 2)

Many expressions are unclear as eg. “phenology is an important index …” page 3  or “Hurst phenomenon” page 9.

In the formula (15) one can delete ai   and - ai and also  bi and - bi , so L=1+t.

The idea of “mutation break points” and the following formula is poorly explained. The text below formula (16) is incomprehensible.

I think that partially the problem with my understanding the manuscript is poor quality of translation eg. instead of “counting variables” there should be “discrete variables”.

The ideas as the necessity of using Geographical Detector Model or Autoregressive model should be introduced before the technicalities of their calculations.

Ad Fig 3. If authors want to analyse differences of WUE for different vegetation types, first differences in NPP and ET for different vegetation types should have been studied.

I am absolutely lost in temporal and spatial resolution of the investigation. In Chapt 3.1.3 there are annual averages, WUESeason, 8dWUESeasones. The whole 3.1.3 chapter is so saturated with numbers that it is difficult to follow. May be some table summarising the most important findings would help the reader to understand  the research results.

Fig. 5 is unclear/incomprehensible. If authors are convinced that there is something important it should be clearly explained.

Fig. 6. What is the meaning of black vertical lines. I cannot find any text commenting this figure. 

Fig.7. This figure also lacks interpretation.

Fig. 8 Why 8b is entitled “future WUE trend”. It is already 2025 so 2018 is the past not the future.

Chapt 3.2.2. “Phenological vegetation parameters are important indicators for detecting vegetation growth. “ Estimation of phenological parameters was not explained. In my opinion Start of vegetation season is very dependent on temperature and this aspect had not been studied here.

Fig 11 is incomprehensible.

Since I do not understand most of the figures I have not tried to review discussion and conclusion subchapters.

Comments on the Quality of English Language

Unfortunately, the document "Spatiotemporal dynamics and driving analysis of water use efficiency in the Mongolian Plateau" is written in a very unclear and chaotic manner. Perhaps it should be split into two shorter texts with clear focuses on the predefined issues, or simply shortened to at most 15 pages. It is a pity that the document does not have line numbering, since it would be more productive to point out passages which need to be corrected.

Some of the formulas are very simple, like the ones for data standardisation (formulas 4, 5, 6) or the Pearson correlation coefficient (formula 22). But some of the statements don't have a proper explanation, like the penalty value with some empirical formula (17).

Also explanation how to calculate linear regression coefficients is definitely redundant (there is however missing second power in the denominator in formula 2)

Many expressions are unclear as eg. “phenology is an important index …” page 3  or “Hurst phenomenon” page 9.

In the formula (15) one can delete ai   and - ai and also  bi and - bi , so L=1+t.

The idea of “mutation break points” and the following formula is poorly explained. The text below formula (16) is incomprehensible.

I think that partially the problem with my understanding the manuscript is poor quality of translation eg. instead of “counting variables” there should be “discrete variables”.

The ideas as the necessity of using Geographical Detector Model or Autoregressive model should be introduced before the technicalities of their calculations.

Ad Fig 3. If authors want to analyse differences of WUE for different vegetation types, first differences in NPP and ET for different vegetation types should have been studied.

I am absolutely lost in temporal and spatial resolution of the investigation. In Chapt 3.1.3 there are annual averages, WUESeason, 8dWUESeasones. The whole 3.1.3 chapter is so saturated with numbers that it is difficult to follow. May be some table summarising the most important findings would help the reader to understand  the research results.

Fig. 5 is unclear/incomprehensible. If authors are convinced that there is something important it should be clearly explained.

Fig. 6. What is the meaning of black vertical lines. I cannot find any text commenting this figure. 

Fig.7. This figure also lacks interpretation.

Fig. 8 Why 8b is entitled “future WUE trend”. It is already 2025 so 2018 is the past not the future.

Chapt 3.2.2. “Phenological vegetation parameters are important indicators for detecting vegetation growth. “ Estimation of phenological parameters was not explained. In my opinion Start of vegetation season is very dependent on temperature and this aspect had not been studied here.

Fig 11 is incomprehensible.

Since I do not understand most of the figures I have not tried to review discussion and conclusion subchapters.

Author Response

Response to Reviewer 3 Comments

Reply

Dear reviewer:

Thank you for your comments on our manuscript entitled” Spatiotemporal dynamics and driving analysis of water use efficiency in the Mongolian Plateau” (ID: sensors-3427641). Those comments are very helpful for revising and improving our paper. We have studied the comments carefully and made corrections which we hope meet with approval. The main corrections are in the manuscript and the responses to the reviewers” comments are as follows (the responses are highlighted in red).

Point 1: Comments and Suggestions for Authors

Unfortunately, the document "Spatiotemporal dynamics and driving analysis of water use efficiency in the Mongolian Plateau" is written in a very unclear and chaotic manner. Perhaps it should be split into two shorter texts with clear focuses on the predefined issues, or simply shortened to at most 15 pages. It is a pity that the document does not have line numbering, since it would be more productive to point out passages which need to be corrected.

Response 1:

We sincerely apologize for this error. Thank you for your careful review. We have made significant and concise revisions to improve the readability of the paper. Line numbers have been added to the article.

Point 2: Some of the formulas are very simple, like the ones for data standardisation (formulas 4, 5, 6) or the Pearson correlation coefficient (formula 22). But some of the statements don't have a proper explanation, like the penalty value with some empirical formula (17).

Response 2:

Thank you for your suggestions. Based on your recommendations, we have appropriately removed some very simple formulas, such as the data normalization formulas (Formulas 4, 5, 6) and the Pearson correlation coefficient formula (Formula 22). We have added some statements with appropriate explanations, and the penalty value is represented by an empirical formula (Formula 17). We have included "Based on the constraint features of the penalty value, a specific penalty function was constructed and added to the objective function, and an unconstrained problem was established using some empirical formulas."

Point 3: Also explanation how to calculate linear regression coefficients is definitely redundant (there is however missing second power in the denominator in formula 2)

Response 3:

We sincerely apologize for this error. Thank you for your careful review. We have thoroughly checked the manuscript and corrected the mistake. Following another reviewer’s suggestion, we have removed the redundant linear regression coefficients.

Point 4: Many expressions are unclear as eg. “phenology is an important index …” page 3 or “Hurst phenomenon” page 9.

Response 4:

Thank you for your suggestions. Based on your recommendations, we have added descriptions related to "phenology is an important indicator..." and the "Hurst phenomenon."

Phenology refers to the timing of natural events, such as the growth and reproduction of plants, animals, and other organisms, either advancing or delaying. It is usually closely related to climate change. Rising temperatures often lead to earlier occurrences of phenomena like plant blooming and animal activity, which affect ecosystems and agricultural production. Phenological changes are an important sign of global warming, reflecting the impact of climate change on the natural world, and have become a key indicator in global climate monitoring.

The Hurst phenomenon refers to the long-term dependence and self-similarity observed in time series data, meaning that past behavior influences future trends. This phenomenon originates from the Hurst exponent, which is used to measure whether a time series exhibits persistence (continuation of trends) or reversibility. A Hurst exponent greater than 0.5 indicates trend persistence, less than 0.5 indicates trend reversal, and equal to 0.5 suggests no memory effect. The Hurst phenomenon is commonly used in the analysis of climate change, financial markets, and other fields.

Point 5: In the formula (15) one can delete ai and - ai and also bi and - bi , so L=1+t.

Response 5:

Thank you for your suggestions. Based on your recommendations, we have also made changes in Formula (15) by deleting ai and - ai as well as bi and - bi so the formula simplifies to L=1+t.

Point 6: The idea of “mutation break points” and the following formula is poorly explained. The text below formula (16) is incomprehensible.

Response 6:

Thank you for your suggestions. We have added the concept of "mutation breakpoint" and the references for the following formula, along with an explanation. A "mutation breakpoint" refers to a point in time series data where a significant change occurs in a variable, typically caused by external shocks or internal mechanisms.

Point 7: I think that partially the problem with my understanding the manuscript is poor quality of translation eg. instead of “counting variables” there should be “discrete variables”.

Response 7:

Thank you for your correction. Based on your suggestion, we have changed "count variable" to "discrete variable." We sincerely apologize for our poor English. We have polished the language through an English language editing service. Additionally, we have corrected some grammatical errors in the manuscript. We apologize again for any inconvenience caused by our poor English.

Point 8: The ideas as the necessity of using Geographical Detector Model or Autoregressive model should be introduced before the technicalities of their calculations.

Response 8 :

Thank you for your correction. Based on your suggestion, we have added an explanation of the necessity of the geographical detector model formula and the autoregressive model before the computational technical details.

The necessity of the geographical detector model formula lies in its ability to quantify the explanatory power of different factors on geographical phenomena, revealing patterns in spatial data. The formula helps identify key factors and spatial heterogeneity by assessing the impact of individual factors and their interactions. It provides a scientific basis for fields such as environmental protection and urban planning. As a quantitative analysis tool, it ensures the accuracy and effectiveness of the analysis, helping to optimize decision-making and resource allocation.

The autoregressive model (AR model) links current data with its past values to reveal dependencies in time series. Its necessity lies in its ability to effectively capture and predict the temporal correlations in data, particularly suited for time series data with long-term trends or seasonal variations. The autoregressive model is widely used in fields such as economics, finance, and meteorology, helping to analyze and forecast future trends and improve the accuracy and reliability of decision-making.

Point 9: Ad Fig 3. If authors want to analyse differences of WUE for different vegetation types, first differences in NPP and ET for different vegetation types should have been studied.

Response 9:

Thank you for your suggestion. We have added supplementary information to the study regarding the differences in NPP and ET across different vegetation types, as shown in Figure 3.

Point 10: I am absolutely lost in temporal and spatial resolution of the investigation. In Chapt 3.1.3 there are annual averages, WUESeason, 8d WUESeasones. The whole 3.1.3 chapter is so saturated with numbers that it is difficult to follow. May be some table summarising the most important findings would help the reader to understand the research results.

Response 10:

Thank you for your suggestion. We have simplified the text description in Section 3.1.3. We used the BFAST method to create time series for the annual average, WUESeason, and 8-day WUESeason, along with the results for the trend component, seasonal component, and residuals. 

Point 11: Fig. 5 is unclear/incomprehensible. If authors are convinced that there is something important it should be clearly explained.

Response 11:

Thank you for your suggestion. We have enlarged the figures in Figure 5 and provided a clear explanation. Figure 5 presents the trend analysis and breakpoint detection of WUESeason, NPP, ET, and 8-day WUESeason during the growing season (April to October) of the Mongolian Plateau using the BFAST model. The results show that WUESeason experienced significant growth between 1997-1998, 2007-2009, and 2009-2018, especially in eastern Mongolia and central Inner Mongolia. The changes in NPPSeason were consistent with WUESeason, while ETSeason significantly decreased during the WUESeason growth periods. This suggests that the short-term sharp increase in WUESeason is associated with the increase in NPPSeason and the decline in ET, which could be driven by human factors.

Breakpoints were detected only in typical grassland and broadleaf forests among different vegetation types. The typical grassland experienced a sharp increase in 1992 (0.36 gC/m²·mm·year, p<0.001) and a sharp decrease in 1999 (0.266 gC/m²·mm·year, p=0.001). No breakpoints were detected in NPPSeason and WUESeason during these two mutation periods, which might have been caused by human disturbances. The broadleaf forest showed significant growth from 1999 to 2017 (0.005 gC/m²·mm·year, p=0.008), but no significant decline was observed between 2017 and 2018 (p>0.05).

Point 12:Fig. 6. What is the meaning of black vertical lines. I cannot find any text commenting this figure.

Response 12:

Thank you for your suggestion. The black vertical lines represent the results of the BFAST model time series for WUESeason on the Mongolian Plateau. The trends for WUESeason are as follows:

From 1982 to 1997, the WUESeason increase trend was 0.02 per year.

From 1997 to 1998, the WUESeason increase trend was 0.135 per year.

From 1998 to 2007, the WUESeason increase trend was 0.002 per year.

From 2007 to 2009, the WUESeason increase trend was 0.121 per year.

From 2009 to 2018, the WUESeason increase trend was 0.09 per year.

Point 13: Fig.7. This figure also lacks interpretation.

Response 13:

Thank you for your suggestion. We have added a textual explanation to the figures in Figure 7. The proportion of pixels showing change points is 15.28% for 1994-1998, 11.98% for 2004-2008, 11.03% for 2009-2013, and 18.04% for 2014-2018.

Point 14: Fig. 8 Why 8b is entitled “future WUE trend”. It is already 2025 so 2018 is the past not the future.

Response 14:

We sincerely apologize for this error and thank you for your careful review. We have removed the word "future" and clarified that 2018 is in the past.

Point 15: Chapt 3.2.2. “Phenological vegetation parameters are important indicators for detecting vegetation growth. “Estimation of phenological parameters was not explained. In my opinion Start of vegetation season is very dependent on temperature and this aspect had not been studied here.

Response 15:

Thank you for the expert's suggestion. In our future research, we will include an analysis of the impact of temperature and WUE data on vegetation phenology in the next phase of our work.

Point 16: Fig 11 is incomprehensible.

Since I do not understand most of the figures I have not tried to review discussion and conclusion subchapters.

Response 16:

Thank you for your suggestion. We used the geographical detector to obtain Figure 11, which analyzes the contribution of factors affecting WUE and the interactions between these factors.

Thank you for your advice, let me harvest a lot. I can't deal with some details well. Please forgive my carelessness. At the same time, your suggestions also let me have a new idea, so that I can better improve the work and sort out the next goal. Thank you again for your guidance.

My English ability is deficient. If some words are not used accurately, you will feel offended. Please forgive me. I will strengthen my English learning in the next study.

If you are dissatisfied, please point out and we will revise your opinion seriously.

I hope that these revisions and the improved text will be satisfactory and make the paper be acceptable for publication in Sensors.

We again very appreciate all your suggestions, comments and favorable consideration.

Sincerely yours,

Yulong Bao Inner Mongolia Normal University.

No. 81, Zhaowuda Road, Hohhot, Inner Mongolia, China

E-mail: baoyulong@imnu.edu.cn

Reviewer 4 Report

Comments and Suggestions for Authors

The article provides a good scientific contribution, integrating a large volume of data on a spatial and temporal scale, combined with statistical methods for analysis and interpretation. Note that the title could include other parameters such as C itself, in addition to the water factor. The abstract should be more conclusive with the results obtained, as well as what and how the results of this article fill a knowledge gap. The introduction is well written and well-founded, as are the materials and methods, written and based on the literature. The results, written and in figures, are clear and concise with the methodology described. The discussion could be better explored, mainly regarding the effects of the increase (temporal scale) of WUE, NPP and ET, for the effects of changes in land use. Explore the relationship of this effect between figures 1 and 2 (of the results). This could present projections and/or scenarios, if land uses are carried out by humanity.

Author Response

Response to Reviewer 4 Comments

Reply

Dear reviewer:

Thank you for your comments on our manuscript entitled” Spatiotemporal dynamics and driving analysis of water use efficiency in the Mongolian Plateau” (ID: sensors-3427641). Those comments are very helpful for revising and improving our paper. We have studied the comments carefully and made corrections which we hope meet with approval. The main corrections are in the manuscript and the responses to the reviewers” comments are as follows (the responses are highlighted in red).

Point 1: Comments and Suggestions for Authors

The article provides a good scientific contribution, integrating a large volume of data on a spatial and temporal scale, combined with statistical methods for analysis and interpretation. Note that the title could include other parameters such as C itself, in addition to the water factor. The abstract should be more conclusive with the results obtained, as well as what and how the results of this article fill a knowledge gap.

Response 1:

Thank you for your suggestion. We have added an analysis based on Net Primary Productivity (NPP), Evapotranspiration (ET), temperature, precipitation, and soil moisture to examine the spatiotemporal variations of WUE in the Mongolian Plateau from 1982 to 2018.

We have adopted your suggestion and changed the title to "Spatiotemporal Variation of Water Use Efficiency in the Mongolian Plateau and Its Driving Factors."

Point 2: The introduction is well written and well-founded, as are the materials and methods, written and based on the literature. The results, written and in figures, are clear and concise with the methodology described. The discussion could be better explored, mainly regarding the effects of the increase (temporal scale) of WUE, NPP and ET, for the effects of changes in land use. Explore the relationship of this effect between figures 1 and 2 (of the results). This could present projections and/or scenarios, if land uses are carried out by humanity.

Response 2:

Thank you for your suggestion. We are pleased to have adopted your advice. We have made additions in the relevant sections.

In the discussion, we have addressed the limitations of the study and future research directions. Currently, due to land use and future scenario data, we have added the impact of land use changes in future scenarios on WUE. We have briefly applied the Hurst index for future analysis, highlighting future research directions.

The water use efficiency (WUE) of terrestrial ecosystems reflects the interaction between vegetation productivity and the availability of water resources. As climate change increases precipitation variability, the infiltration of rainfall directly affects the replenishment of soil moisture. These factors significantly influence the ecosystem's evapotranspiration process, which in turn affects vegetation growth and the water-carbon cycle, leading to changes in WUE across different vegetation types. Future warming will further intensify the supply-demand conflict of water use in terrestrial ecosystems. Therefore, revealing the changes in water use efficiency under future warming scenarios is crucial for ensuring the sustainable use of water resources. Future studies should use indicators such as rainfall, soil moisture, and evapotranspiration under 1.5℃ and 2.0℃ warming scenarios to further understand the impact of climate change on ecosystems in the arid and semi-arid regions of the Mongolian Plateau.

Thank you for your advice, let me harvest a lot. I can't deal with some details well. Please forgive my carelessness. At the same time, your suggestions also let me have a new idea, so that I can better improve the work and sort out the next goal. Thank you again for your guidance.

My English ability is deficient. If some words are not used accurately, you will feel offended. Please forgive me. I will strengthen my English learning in the next study.

If you are dissatisfied, please point out and we will revise your opinion seriously.

I hope that these revisions and the improved text will be satisfactory and make the paper be acceptable for publication in Sensors.

We again very appreciate all your suggestions, comments and favorable consideration.

Sincerely yours,

Yulong Bao Inner Mongolia Normal University.

No. 81, Zhaowuda Road, Hohhot, Inner Mongolia, China

E-mail: baoyulong@imnu.edu.cn

Round 2

Reviewer 1 Report

Comments and Suggestions for Authors

I appreciate the changes made to the manuscript. I believe the authors have adequately addressed the observations, and the current version has significantly improved. I have no further comments and agree with its publication.

Author Response

We sincerely thank the reviewers for their detailed review and valuable suggestions on our manuscript. The feedback provided by the reviewers has greatly helped us improve our research and make our manuscript more complete. We are very grateful for the constructive feedback offered by the reviewers, as it has significantly enhanced our work. Once again, we appreciate the reviewers' support and assistance with our research.

Reviewer 3 Report

Comments and Suggestions for Authors

Dear Authors,

Unfortunately I have not received any specific documentation with answers to my remarks.

The latest version of the document "Spatiotemporal dynamics and driving analysis of water use efficiency in the Mongolian Plateau" is written more clearly and with proper English. However  still I am on opinion that it is too long .

There are also some detailed comments:

Line 44: Instead of “ Owing to the greenhouse effect, progress of civilization,and intensification of human activities” I suggest “Owing to progress of civilization, intensification of human activity thus speeding up green house effect,”

Formula (1)- There is no need to present this formula since it would be enough to say that the data were normalised.

Line 227 “that outliers are robust to a certain extent” - the method can be robust to outliers  not the other way round

|Z|>u1 - a /2 - there is something wrong with this expression

Line 270- 271 : not clear

In the subchapter about Hurst index  there is no explanation of the R/S – meaning

Also the sentence below formula (14) is not clear to me.

Pearson coefficient  is so well known coefficient that this subchapter  Pearson correlation coefficient is redundant.

Line 354: some reference is needed for application of AR models .

I also maintain my suggestion that the Results section should include a table summarizing the key findings to help the reader understand the study results.

Figure 10b) I would prefer to have it in the standard Heat map mode where the correlation matrix is symmetric.

Kind regards,

The reviewer

Author Response

Response to Reviewer 3 Comments

Reply

Dear reviewer:

Thank you for your comments on our manuscript entitled” Spatiotemporal dynamics and driving analysis of water use efficiency in the Mongolian Plateau” (ID: sensors-3427641). Those comments are very helpful for revising and improving our paper. We have studied the comments carefully and made corrections which we hope meet with approval. The main corrections are in the manuscript and the responses to the reviewers” comments are as follows (the responses are highlighted in red).

Point 1: The latest version of the document "Spatiotemporal dynamics and driving analysis of water use efficiency in the Mongolian Plateau" is written more clearly and with proper English. However  still I am on opinion that it is too long .

Response 1:

Thank you for your suggestion. You are correct. We have moved the description of some methods, results, and relevant figures to the appendix, which has reduced the content in the main text.

Point 2: Line 44: Instead of “ Owing to the greenhouse effect, progress of civilization,and intensification of human activities” I suggest “Owing to progress of civilization, intensification of human activity thus speeding up green house effect,”

Response 2:

Thank you for your suggestion. We have revised the manuscript based on your advice. (P2Lines 44—45)

Point 3: Formula (1)- There is no need to present this formula since it would be enough to say that the data were normalised.

Response 3:

Thank you for your suggestion. Based on your advice, we have removed Equation (1) from the manuscript.

After the revision:

Drought severity indices (DSI) can provide more information on water stress [59]. The DSI was estimated using extended GIMMS MODIS NDVI, AVHRR ET, and ERA5-LAND PET data over 37 years.

2.2.4. WUE dataset

The WUE was calculated as the ratio of GLASS NPP to AVHRR ET. Since the focus of this study was land water use efficiency in vegetation areas, we removed non-vegetation areas (average NDVI<0.1), ET, and NPP before the inversion of water use efficiency. Therefore, the final WUE also excluded data from non-vegetated areas.(P5Lines 183—190)

Point 4: Line 227 “that outliers are robust to a certain extent” - the method can be robust to outliers  not the other way round

Response 4:

Thank you for your suggestion. We have made the replacement as per your advice, and this section has been moved to the appendix.

After the revision:

If WUE is positive, it means that WUE is increasing; if it is negative, WUE is decreasing. The Mann–Kendall test was used to determine whether a trend was significant. Similarly, it considers that the sample data may not obey a certain distribution, and the method can be robust to outliers [64,65].

Point 5: |Z|>u1 - a /2 - there is something wrong with this expression

Response 5:

Thank you for your suggestion and for your careful review. We have corrected this error, and this section has been moved to the appendix.

After the revision:

After determining the significance level α under the conditions of | Z | > u1–α/2, the dynamics of the trend were determined to be significant.  

Point 6: Line 270- 271 : not clear

Response 6:

Thank you for your suggestion. We have removed some redundant information to make it easier to read.

After the revision:

The change point detection tool in ArcGIS pro3.0 software was used to detect the change points of the annual WUE. This tool enables us to check changes in the slope (linear trend) of discrete variables. Change-point detection is relatively similar to time-series anomaly detection but differs in several important aspects.(P7Lines 240—243)

Point 7: In the subchapter about Hurst index  there is no explanation of the R/S – meaning

Response 7:

Thank you for your suggestion. We have added an explanation of the meaning of R/S in the manuscript as per your advice.

After the revision:

The Hurst index (H) is used in prediction studies across demographics, economics, and climate change. R/S analysis helps analyze the historical memory and fractal characteristics of time series. R/S stands for Rescaled Range analysis, which assesses a time series' volatility and long-term dependence by comparing the range to the standard deviation. (P7Lines265—269)

Point 8: Also the sentence below formula (14) is not clear to me.

Response 8 :

Thank you for your suggestion. We have revised the description based on your advice to improve its readability.

After the revision:

To confirm the presence of the Hurst phenomenon, the condition r must satisfy the R/S criteria. The Hurst index (H) was calculated using least-squares regression: log(R/S) n = a + H × log(n). The H values indicate whether the WUE time series is continuous or random. When H is between 0 and 0.5, the trend is unsustainable, meaning future trends reverse past ones. When H is between 0.5 and 1, the trend is persistent, with future trends aligning with past ones. An H value of 0.5 suggests a random WUE trend.

In our study, annual WUE served as the input for the model. We applied Theil–Sen median trend analysis and the Mann–Kendall test to quantify significant trends in water use efficiency. The resulting Hurst index was incorporated into the analysis of significant WUE differences, and spatial differentiation of future WUE trends was predicted.(P8Lines280—291)

Point 9: Pearson coefficient  is so well known coefficient that this subchapter  Pearson correlation coefficient is redundant.

Response 9:

Thank you for your suggestion. We have removed the content related to the Pearson correlation coefficient from the manuscript based on your advice.

Point 10: Line 354: some reference is needed for application of AR models .

Response 10:

Thank you for your suggestion. We have updated the relevant literature on the AR model based on your advice. 

After the revision:

  1.     Mabrouk, A.B.; Abdallah N.B.; Dfifaoui, Z. Wavelet decomposition and autoregressive model for time series prediction, Appl Math Comput, 2008, 199(1), 334-340.(P28Lines880—881)
  2.     Kaur, J., Parmar, K.S. & Singh, S. Autoregressive models in environmental forecasting time series: a theoretical and application review. Environ Sci Pollut Res, 2023, 30, 19617–19641.(P28Lines880—881)

Point 11: I also maintain my suggestion that the Results section should include a table summarizing the key findings to help the reader understand the study results.

Response 11:

Thank you for your suggestion. It’s a great idea. We have added a summary description at the end of the results section, which contains more detailed and complex content, to help readers more intuitively understand these findings.

After the revision:

Overall, from 1982 to 2018, NPP decreased in northern and northeastern Mongolia, while WUE increased in central and southern regions. The highest WUE variability occurred in desert grasslands.(P10Lines371—373)

Overall, WUESeason on the Mongolian Plateau showed significant increases in 1997-1998, 2007-2009, and 2009-2018, particularly in eastern Mongolia and central Inner Mongolia, linked to higher NPPSeason and lower ET.(P12Lines401—403)

Overall, PRE and SW were positively correlated with WUE in most areas of the Mongolian Plateau, while TEM and DSI were negatively correlated with WUE, particularly in the central and eastern regions. Water loss and increased temperature led to decreased WUE, especially in arid and semi-arid areas. SOS showed a positive correlation with WUE, NPP, and ET, with delayed SOS improving WUE and NPP, while EOS had minimal effect on WUE.(P16Lines484—489)

Overall, WUE on the Mongolian Plateau shows strong resilience to water anomalies but weak resistance to temperature, drought, and vegetation greenness anomalies, with forest ecosystems being more sensitive to temperature and drought disturbances.(P18Lines539—542)

Point 12:Figure 10b) I would prefer to have it in the standard Heat map mode where the correlation matrix is symmetric.

Response 12:

Thank you for your suggestion. However, this section uses the plotting tool provided by the geographic detector module, which includes different forms of relationships such as nonlinear and linear patterns, making it potentially more useful. Heatmaps are a great way to visualize correlations, but they may not convey as much information. After careful consideration, we have decided to retain the original plot.

Thank you for your advice, let me harvest a lot. I can't deal with some details well. Please forgive my carelessness. At the same time, your suggestions also let me have a new idea, so that I can better improve the work and sort out the next goal. Thank you again for your guidance.

My English ability is deficient. If some words are not used accurately, you will feel offended. Please forgive me. I will strengthen my English learning in the next study.

If you are dissatisfied, please point out and we will revise your opinion seriously.

I hope that these revisions and the improved text will be satisfactory and make the paper be acceptable for publication in Sensors.

We again very appreciate all your suggestions, comments and favorable consideration.

Sincerely yours,

Yulong Bao Inner Mongolia Normal University.

No. 81, Zhaowuda Road, Hohhot, Inner Mongolia, China

E-mail: baoyulong@imnu.edu.cn
